# The Bladder Tumor Microenvironment Components That Modulate the Tumor and Impact Therapy

**DOI:** 10.3390/ijms241512311

**Published:** 2023-08-01

**Authors:** Mugdha Vijay Patwardhan, Ratha Mahendran

**Affiliations:** Department of Surgery, Yong Loo Lin School of Medicine, National University of Singapore, Singapore 119228, Singapore; mugdha.p@u.nus.edu

**Keywords:** bladder cancer, tumor microenvironment, single cells

## Abstract

The tumor microenvironment (TME) is complex and involves many different cell types that seemingly work together in helping cancer cells evade immune monitoring and survive therapy. The advent of single-cell sequencing has greatly increased our knowledge of the cell types present in the tumor microenvironment and their role in the developing cancer. This, coupled with clinical data showing that cancer development and the response to therapy may be influenced by drugs that indirectly influence the tumor environment, highlights the need to better understand how the cells present in the TME work together. This review looks at the different cell types (cancer cells, cancer stem cells, endothelial cells, pericytes, adipose cells, cancer-associated fibroblasts, and neuronal cells) in the bladder tumor microenvironment. Their impact on immune activation and on shaping the microenvironment are discussed as well as the effects of hypertensive drugs and anesthetics on bladder cancer.

## 1. Introduction

Cancer likely develops from aberrant cells as a consequence of exposure to chemical or biological damage that gives them a proliferative advantage. The cancer cells shape their surroundings, the tumor microenvironment (TME), in such a manner as to ensure it is favorable for their growth, proliferation, and survival. The TME is complex and contains a number of different cell types (Figure 1) that all work under the direction of the cancer cells, although the non-cancerous cells in the TME also influence the cancer cells, leading to the ultimate complexity of the TME (Table 1). Bladder cancer (BC) cells have been shown to influence the TME by secreting various soluble factors as well as cargo-carrying extracellular vesicles, which impact the cells listed in Table 1. Given this complexity, it is not surprising that therapy that only targets the cancer cells may not be sufficient to cure cancer. 

Non-muscle invasive bladder cancer (NMIBC) is characterized by frequent recurrences and patients require life-long monitoring, making it one of the more expensive cancers to manage [1]. There are few biomarkers that have been confirmed to be predictive of patient response to therapy. The standard of care for NMIBC is transurethral resection of the tumor (TURBT) followed by Mycobacterium bovis Bacillus Calmette Guérin (BCG) immunotherapy [2]. However, some 30–50% of patients fail this therapy and newer therapies are more expensive but have only incremental benefits. NMIBC can progress to muscle invasive disease (MIBC) and metastasize to other tissues. For high-grade, high-risk NMIBC and MIBC, radical cystectomy is performed to remove the bladder [2]. A better understanding of the TME may provide insights into the development and recurrence of BC and may identify new targets for improving therapeutic response.

The tumor microenvironment is highly dynamic, consisting of numerous cell types, including cancer cells, stem cells, immune cells such as neutrophils, macrophages, DCs, MDSCs, T-cells, B-cells, fibroblasts, adipocytes, endothelial cells, pericytes, neurons, and neuro-endocrine cells. These cells interact with one another and influence the local tumor environment. The balance between immune cell types as well as their functional activation plays an important role in tumor prognosis and their activation is influenced by the TME. 

## 2. Bladder Cancer Cells

### 2.1. Soluble Factors

Tumor-derived soluble factors aid in establishing a pro-inflammatory and angiogenic TME (Table 2), which promotes cancer cell proliferation and metastasis [3]. Interleukin-1 (IL-1), secreted by T24 bladder cancer cells, was found to induce endothelial activation and angiogenesis (vessel formation) in human umbilical vein endothelial cells (HUVEC) cells [4]. T24 supernatants promoted the breakdown of the endothelial monolayer, resulting in the formation of leaky vasculature, and this could be blocked with IL-1 receptor antagonists (IL-1ra) [4]. Interestingly, analysis of human BC tissue biopsies revealed low levels of IL-1ra compared to the healthy urothelium, suggesting that cancer cells may create imbalances in the ratio of IL-1 and IL-1ra to promote the formation of tumor vasculature. Proteomic analysis of the conditioned medium from RT4 (BC) cells identified IL-8, vascular endothelial growth factor-A (VEGF-A), and tissue inhibitor of metalloproteinase-1 (TIMP-1), which may have contributed to HUVEC activation, and this was validated in vitro by blocking either IL-8 or VEGF-A, which dampened HUVEC activation [5]. In a separate study, knockdown of VEGF in patient-derived BC cells led to decreased migration and invasion (matrigel and transwell assays) of BC cells [6]. Studies with the human BC cell lines HT-1376, T24, 5637, and J82 have revealed a potential role for long non-coding RNAs (lncRNA) in modulating the TME. Tumor-derived long intergenic non-coding RNA-482 (LINC00482) was shown to increase levels of pro-inflammatory cytokines such as IL-6, tumor necrosis factor alpha (TNFα), and IL-1 beta (IL-1β), as well as VEGF, which led to increased inflammation and angiogenesis [7]. *PKM2* gene ablation in cancer cells led to reduced growth and apoptosis and in a H-ras-driven urothelial carcinoma model in mice, it was shown to control angiogenesis via regulation of hypoxia inducible factor 1 alpha (*HIF1α*) and *VEGF* expression [8]. 

Cancer cells secrete cytokines/chemokines, such as IL-10, transforming growth factor beta (TGF-β), and chemokine (C-C motif) ligand 2 (CCL2), which exert immune-suppressive effects and promote the recruitment and activation of suppressive cells. The murine bladder cancer cell line MB49 has been shown to secrete IL-10, which impairs dendritic cell activation and maturation, and in turn results in dampened activation of cytotoxic T-cells [9]. MB49 cells also secrete CCL2, which promotes M2 polarization of tumor associated macrophages (TAM) after stimulation with radiation [10]. Additionally, bone morphogenetic protein 4 (BMP4) from UMUC3 (BC) cells was also found to induce M2 polarization of macrophages [11]. The human BC cell lines RT112 and Cal-29 have been shown to secrete cytokines (IL-1 and IL-8), which recruit stromal cells such as fibroblasts to the TME. These fibroblasts further promote cancer invasion and migration [12]. A study by Miyake et al. found that BC cell-derived CXCL1 (from UMUC3, T24, and J82 BC cell lines) led to the recruitment and activation of both TAMs and fibroblasts [13]. Mouse BC cells (MBT-2) have also been shown to increase the expression of the immunosuppressive marker programmed death ligand-1 (PD-L1) in co-cultured bone-marrow derived macrophages and myeloid-derived suppressor cells (MDSCs), thus reducing the activation of splenic T-cells and the anti-tumor response [14]. Additionally, loss of the Y-chromosome in BC cells may also contribute to their ability to influence the TME. Implantation of MB49 cells (lacking the Y-chromosome) into mice resulted in an increase in the infiltration of immunosuppressive macrophages (PD-L1+) [15]. Such tumors may be more responsive to immune checkpoint therapy targeting the programmed cell death protein-1 (PD-1)/PD-L1 axis.

**Table 2 ijms-24-12311-t002:** Cancer cell modulation of the TME.

Study	Cell Line	Factors	Impact	Reference
Mouse models				
In vivo mouse model	MB49	IL10	Impairs dendritic cell activation and maturation Dampens activation of cytotoxic T-cells	[9]
In vitro allograft	MB49	CCL2	Induces M2 polarisation of tumor-associated macrophages (TAMs)	[10]
In vivo mouse model	-	Pyruvate kinase M2 (PKM2) controlled vascular endothelial growth factor (VEGF)	Increased angiogenesis	[8]
In vivo mouse model	MBT-2	Prostaglandin E 2 (PGE_2_)/cyclooxygenase 2 (COX2)	Increased expression of PD-L1 on macrophages and myeloid-derived suppressor cells (MDSC)	[14]
Human cell lines and cancer databases				
In vitro assays (Boyden Chamber)	RT112, Cal-29	IL-8, IL-1	Recruit fibroblasts to the TME Paracrine signaling with fibroblasts, which promotes cancer invasion and migration	[12]
In vitro assay	T24	IL-1 and IL-1α	Endothelial activation Formation of leaky tumor vasculature	[4]
In vitro assay	Various Cell Lines	BMP4 secretion	Induces differentiation of M2 macrophages	[11]
In vitro and microfluidic assays	RT4, RT112, and T24/83	VEGF-A, IL-8, TIMP1	Endothelial activation	[5]
Human Database	The Cancer Genome Atlas (TCGA)	CCL4	Impaired T-cell activation due to decreased recruitment of DCs	[16]
In vitro assay	Patient derived Cell line	VEGF, cylin dependant kinase 4 (CDK4)	Increased migration and metastasis Increased angiogenesis	[6]
In vitro (transwell) assays	T24 253J	VEGF, CXC motif chemokine ligand 1 (CXCL1), CXCL5, CXCL8	Recruitment of endothelial cells (angiogenesis) Endothelial activation	[17]
In vitro assays	HT-1376, T24,5637(HTB-9), and J82	LINC00482	Increased inflammation (increased levels of IL-6, TNF, and IL-1β) Increased angiogenesis (VEGF)	[7]
3D cell culture and in vitro assays	J82, UMUC3, T24	CXCL1	Recruitment and activation of tumor-associated macrophages and cancer-associated fibroblasts	[13]
In vitro assays	T24 and UMUC3	Brain-derived neurotrophic factor (BDNF)	Paracrine signaling to BC cells Increased proliferation and invasion of BC cells	[18]

### 2.2. Extracellular Vesicles (EVs)

EVs are membrane-bound structures that are released by cells to mediate cell-to-cell communication and extracellular transport [19]. These structures range from 50 to 1000 nanometers (nm) in diameter, may be derived from the endosomal system (exosomes) or plasma membrane (microvesicles), and can carry various cargo that can then influence processes such as proliferation, migration, and the immune response in surrounding cells and distant locations. In the TME, tumor-derived EVs have been shown to influence the surrounding environment and tumor-infiltrating cells (Table 3). EVs, isolated from T24 and TCCSUP BC cells, increased the migration and invasion of 5637 cells in transwell assays and increased HUVEC activation in terms of tube formation [20]. Protein analysis of the BC-derived exosomes revealed high levels of proteins known as endothelial locus 1 (DEL1) or epidermal growth factor-like repeat and discoidin 1-like domain 3 (EDIL-3). Silencing *EDIL-3* decreased the pro-tumorigenic effects of the EVs. Recently, lncRNAs, contained within BC-derived EVs, have shown diverse effects on BC cells themselves as well as the surrounding TME. Exposure of UMUC3 cells to EVs from 5637 cells increased their viability and promoted migration. Quantitative real-time PCR analysis detected the high expression of the lncRNA urothelial carcinoma-associated 1 (*UCA-1*) in the EVs. Small hairpin RNA (shRNA)-induced inhibition of *UCA-1* reduced the EV-driven increase in the invasive capacity of UMUC3 cells. Treatment of 5637 cells with the lncRNA-*UCA1* shRNA led to decreased expression of the epithelial-mesenchymal transition (EMT) markers vimentin and metalloproteinase 9 (MMP9) [21]. EMT is a naturally occurring process through which epithelial cells acquire the characteristics of mesenchymal cells that are associated with increased migration [22]. However, in tumors, cancer cells that undergo EMT display an invasive phenotype that invades surrounding tissue and migrates to distant locations (metastasis) [22]. In BC, EMT leads to tumor progression, especially from NMIBC to MIBC; thus, the acquisition of EMT markers is indicative of the potential for tumor progression [23]. Additionally, the lncRNAs *Hox antisense intergenic RNA* (*HOTAIR*) [24], lymph node metastasis-associated transcript 2 (LN*MAT2*) [25], LINC00960, and LINC02470 [26], all contained within EVs from BC cells (various cell lines), have all been shown to promote cancer cell migration, invasion, and EMT. Interestingly, tumor-derived EVs from MB49 cells have also been shown to influence the development of cancer stem cells by promoting their self-renewal and enhancing the expression of genes associated with stem-like properties [27]. 

Additionally, tumor-derived EVs can also influence the non-cancer cells within the TME. EVs from bladder cancer cell lines T24 and RT4 were found to induce the transformation of healthy fibroblasts (isolated from bladder tissue) into cancer-associated fibroblasts (CAF), measured by an increase in the levels of CAF-associated proteins α smooth muscle actin (αSMA), fibroblast-activated protein (FAP), and galectin [28]. Additionally, fibroblasts that were exposed to the EVs also displayed increased proliferation capacity and TGF-β production [28]. A study by Huyan et al. highlighted the effect of BC-derived EVs on natural killer (NK) cells. In vitro studies revealed that microRNA-221-5p (miR-221-5p) and miR-186-5p in EVs purified from T24 cells decreased NK cell viability through the upregulation of pro-apoptotic pathways and decreased their cytotoxic ability, thus impairing their anti-tumorigenic function [29]. 

Cancer cells (within primary tumors) are also able to influence potential metastasis sites by forming a pre-metastatic niche (PMN) through EV-induced recruitment of bone marrow-derived hematopoietic progenitor cells [30]. However, this has not been well-studied in the context of bladder cancer. A study by Silvers et al. compared the expression of tenascin-C (TNC), an extracellular matrix glycoprotein involved in tissue remodeling, in benign lymph nodes of BC patients and found high TNC levels to be associated with metastasis, suggesting that it could be a marker of the pre-metastatic niche in BC [31]. EVs from J82 and TCCSUP BC cells were also found to induce TNC expression in primary fibroblasts, resulting in their activation. Thus, it was hypothesized that such BC-derived EVs may travel to pre-metastatic sites and induce the expression of TNC, but this needs to be verified [31]. Cancer cells secrete exosomes containing ephrin B1, a protein involved in embryogenesis, axon guidance, and angiogenesis. Ephrin B1 binds to Eph receptor tyrosine kinase and potentiates tumor growth [32]. Ephrin B silencing in invasive MB49 bladder cancer cells (MB49-I) resulted in lower migratory and invasive capacity [33].

**Table 3 ijms-24-12311-t003:** The impact of tumor-derived EVs on the TME.

Study	Cell Line	Factors	Impact	Reference
Mouse				
In vitro (sphere formation, migration and invasion assays)	MB49 Cells	Unidentified factor	Self-renewal of cancer cells Increased stem-like properties in surrounding cancer cells	[27]
In vivo implantation in nude mice	T24	Cathepsin B	Increased angiogenesis	[34]
In vivo implantation in BALB/c nude mice	T24/5637	Circular RNA PRMT5 (circPRMT5)	Increased lung metastasis	[35]
Human				
In vitro cell co-culture	T24 and RT4	TGF-β	Transformation of healthy bladder-resident fibroblasts into cancer-associated fibroblasts	[28]
In vitro assays (wound-healing and tube-formation assays)	TCCSUP, T24	EDIL-3	Facilitates tumor cell migration and invasion Promotes angiogenesis	[20]
In vitro assays (transwell and migration)	TCCSUP, T24	*HOTAIR*	Promotes migration, invasion, and EMT of bladder cancer cells	[24]
Various in vitro assays	TSGH-8301, T24, J82	LINC00960 and LINC02470	Increased tumor cell viability, invasion, and clonogenicity Promotes EMT	[26]
In vitro co-culture and assays	T24, SV-HUC-1	miR-221-5p and miR-186-5p	Impaired NK cell function and viability (increased apoptosis of NK cells) Reduced cytotoxic activity of NK cells	[29]
In vitro assays	T24	MiR-217	Blocks ferroptosis in cancer cells	[36]
In vitro co-culture and assays	5637	lncRNA-*UCA1*	Promotes cancer cell growth and invasion Increased EMT	[21]
In vitro assays	T24	Unidentified factor	Decreased tumor cell apoptosis through upregulation of PI3K/Akt and ERK pathways	[37]
In vitro co-culture and assays	T24 and UMUC3	Unidentified factor	Increased migration, invasion, and EMT	[38]
In vitro assays	T24, TCCSUP, 5637 and UMUC3	circPRMT5	Increased tumor aggressiveness Increased EMT	[35]
In vitro co-culture and assays	5637 and UMUC3	LN*MAT2*	Promotes lymphangiogenesis and metastasis	[25]
In vitro migration and tube-formation assays	5637 and UMUC3	Brain cytoplasmic RNA 1 (*BCYRN1*)	Promotes lymphangiogenesis and metastasis through Wnt signaling	[39]

### 2.3. Bladder Cancer Stem Cells

Cancer stem cells (CSCs) are cancer cells that display stem-like properties, such as the capacity for self-renewal, differentiation into multiple cancer lineages, and the ability to establish tumors that are heterogenous in nature [40]. They are able to rapidly form spheres in in vitro culture and can form tumors even upon transplantation of a small number of CSCs into mice. Many studies have shed light on the ability of CSCs to maintain cancer cell differentiation, growth, and heterogeneity. Within the TME, CSCs reside in niche environments, specifically localizing in the innermost oxygen-deprived regions where the expression of hypoxia-related genes promotes the maintenance of their stem-like characteristics [41]. 

The healthy bladder consists of urothelial stem cells, found in the basal layer, which aid in replenishing populations of damaged or dead urothelial cells [42]. Bladder CSCs may originate from urothelial stem cells upon the acquisition of mutations or alternatively, mutations in non-stem urothelial cells may also result in CSCs. Studies have shown that bladder CSCs are positive for basal cell markers such as CD44 and cytokeratin 5 (CK5), which may indicate that CSCs originate from the basal layer, perhaps from urothelial stem cells [43]. Additionally, bladder CSCs express markers such as those involved in self-renewal (CD44, pentaspan transmembrane glycoprotein, CD133, and aldehyde dehydrogenase 1 (ALDH1)); proliferation and cancer progression (SRY homology box 2 (SOX2)); tumor initiation (histone 3 lysine 4 methyl transferase 2 (MLL2), AT rich interactive domain-containing protein 1A (ARID1A), and G protein-coupled receptor family C group 5 member A (GPRC5A)); as well as increased mutations and dysregulated activity of genes involved in tumor progression and EMT (sonic hedgehog protein (Shh), Wnt/B-catenin, phosphoinositide 3-kinase (PI3K)/Akt, neurogenic locus notch homology protein 1 (Notch1), and TGF-β) [44]. Introduction of co-mutations in a selection of these genes in BC cells, such as MLL2, ARID1A, CREB-binding protein (CREBBP), or GPRC5A, resulted in an increase in the stem-like properties of these cells [45]. Namely, these cells were better able to form spheres (a property of CSCs) in vitro and initiate bladder tumors in vivo. Common pathways that are highly active in both cancer cells and CSCs, such as the PI3K-AKT pathway, can also fuel positive feedback loops between tumor cells that facilitates a pro-tumorigenic TME [46]. 

Whilst not specific to bladder cancer, the CSC secretome and its functions have been well reviewed by López de Andrés et al. [47]. CSCs recruit both stromal and immune cells to the TME through the secretion of various soluble factors and/or micro-vesicles [47]. Single-cell sequencing analysis of tumor tissues from bladder cancer patients with low and high risk of recurrence and recurrent BC revealed that CSCs were enriched in patients with recurrences [48]. Two distinct populations of bladder CSCs were identified: one with higher expression of lysine demethylase 5B (KDM5B), which promotes cell cycle progression through activation of the E2F/RB pathway; and the other with increased expression of enhancer of zeste homolog 2 (EZH2, a histone lysine N-methyltransferase), which promotes EMT through suppression of NCAM1 [48]. Thus, CSCs are heterogenous and some may be involved in cancer recurrence [48,49]. Single-cell analysis of human BC tissues (sourced from public database) identified a population of CSCs that were ‘quiescent’ and had lower expression of genes involved in pathways such as cell division, chromosome segregation, and DNA binding compared to proliferative CSCs [50]. Dormant CSCs may escape destruction by chemo and radiation therapies that target fast-dividing cells, which could be the cause of bladder cancer recurrence and metastasis. Evidence for the generation of dormant CSCs comes from studies of dormancy induction in T24 or MB49 cells in vivo upon exposure to CD8 T-cell-derived IFN-γ, which activates the indolamine-2, 3-dioxygenase/kynurenine/aryl hydrocarbon receptor/P27 (IDO/Kyn/AHR/P27) signaling pathway [51]. Mice with orthotopic MB49 tumors treated with IDO inhibitors showed better prognosis and higher survival [51]. Thus, the targeting of bladder CSCs is an important treatment approach; however, the direct targeting of CSCs has been challenging due to their similarities with normal stem cells [52].

### 2.4. Targeting CSCs

Strategies involving the application of tumor vaccines, such as a streptavidin-mouse GM-CSF combined with murine MB49 bladder CSC-based vaccine, have been shown to effectively sensitize immune cells against both cancer cells and CSCs [53]. Other methods involve targeting specific molecules or pathways that are highly expressed in CSCs, such as heat-shock proteins (Hsp90), CD47, TGF-β, the COX2-PGE2 axis, or the Wnt signaling pathway [42]. Hsp90 is a molecular chaperone that has been shown to support the stability of cytoplasmic proteins involved in pro-tumorigenic pathways and in CSCs. Hsp90 is associated with increased self-renewal and proliferation capacity [54]. In vitro studies and a xenograft model showed that the application of a Hsp90 inhibitor increased targeting of CSCs, thus reducing resistance to cisplatin therapy [54]. The COX2-PGE2 pathway is associated with CSC repopulation and oncogenesis in BC; targeting of the COX2/PGE2 axis through application of the anti-diabetic drug metformin decreased the expression of stem cell markers (CK14 and transcription factors OCT3/4) in mouse bladder tumors [55]. Another xenograft model showed improved prognosis in response to chemotherapy when combined with an anti-PGE2 monoclonal antibody and a COX2 inhibitor [56]. Mechanisms aimed at targeting CD47, TGF-β, and Wnt have had limited success in pre-clinical work and studies are still underway to explore targeting these pathways in BC [42]. 

## 3. Cancer-Associated Fibroblasts (CAFs)

CAFs are the most abundant stromal cells in the TME and there is cross-talk between tumor cells and CAFs. Single-cell profiling of tumors from BC patients revealed the presence of CAFs that could be clustered into two main categories: inflammatory CAFs (iCAF) and myo-CAFs (mCAFs) [57]. ICAFs presented with higher expression of inflammatory cytokines and chemokines, including CXCL12, IL6, CXCL14, CXCL1, and CXCL2, whereas mCAFs were found to express genes regulating the extracellular matrix (ECM), focal adhesion, and the muscle system [57]. Pathway analysis further revealed that iCAFs were involved in degradation of the ECM and recruitment of blood vessels [57]. In vitro studies revealed an increased proliferation and migration capacity of T24 bladder cancer cells when co-cultured with iCAFs [57]. 

Conditioned medium (CM) from CAFs isolated from patient bladder tumors was able to increase the proliferation and invasion of T24 and J82 cells. This effect was abrogated upon application of an anti-IL-1B antibody [58]. Analysis of differentially expressed genes in BC cells revealed upregulation of Wnt target genes upon exposure to iCAF CM in an IL-1B-dependent manner [58]. The Wnt signaling pathway is active in many cancers and has been shown to regulate tumorigenesis (especially EMT), the stem-like phenotype, and resistance to therapy [59]. Activation of the frizzled receptor by Wnt leads to the activation and nuclear translocation of β-catenin, which in turn results in the transcription of numerous genes including oncogenes (c-myc, fibroblast growth factor (FGF), and epidermal growth factor (EGF)), cell cycle genes (cyclin-D1), and stem-like phenotype-related genes (CD44 and ALDH1) [60]. Another study isolated a cluster of CAFs that expressed the membrane-located urea transporter SLC14A1. These SLC14A1^+^ CAFs were shown to increase the stem-like characteristics of BC cells (T24, UMUC3, and SYBC1), as measured by an increased number of cells expressing the stem cell markers ALDH1 and CD133 [61]. When tested in a xenograft model, introduction of SLC14A1^+^ CAFs into the tumor increased tumor growth [61]. Knockdown of Wnt5A in SLC14A1^+^ CAFs resulted in decreased expression of CD133 in BC cells [61]. Both IL-6 and CCL2 expression in CAFs was found to be associated with increased invasion of T24 bladder cancer cells [62]. Interestingly, introduction of the estrogen receptor ER-a into CAFs further increased the expression of IL-6 and CCL2, which in turn increased CAF-induced BC cell invasion [62]. CAFs isolated from human bladders were shown to decrease the expression of epithelial markers (E-cadherin, phospho-β-catenin, and phospho-glycogen synthase kinase 3β (phospho-GSK3β)) and increase the expression of mesenchymal markers (N-cadherin and vimentin) and EMT-related transcription factors (SNAI1, ZEB1, and TWIST1) in RT4 cells in an IL-6-dependent manner [63]. CAF-derived TGF-β in BC was associated with tumor metastasis (EMT) and an immunosuppressive TME. Comparison of CAFs and healthy fibroblasts isolated from human bladder tumors and adjacent tissues, respectively, revealed that CAFs had significantly high expression of TGF-β [64]. Conditioned medium from CAFs was able to increase the expression of EMT-related genes in three BC cell lines (T24, 5637, and J82), the effect of which was diminished upon blocking TGF-β [64]. 

CAFs also mediate their effect on the TME through secretion of EVs. CAFs isolated from BC tissue were found to deliver miRNA-146a-5p through exosomes to T24 and 5637 cells, which led to an increase in signal transducers and activators of transcription (STAT) signaling and stem-like characteristics [65]. CAF-derived exosomal LINC00355 was found to be transferred to BC cells (T24 and 5637), promoting proliferation and invasion [66] as well as facilitating cisplatin resistance through upregulation of the miR-34b-5p/ ATP binding cassette (ABC) subfamily B member 1 (ABCB1) axis [67]. Apart from their influence on BC cells, CAFs have also been shown to influence tumor-infiltrating immune cells. Co-culture of T24 and CD8^+^ T-cells with EVs from primary human BC-associated CAFs in a transwell double chamber-based system showed a decrease in T-cell proliferation, measured by a decrease in the level of T-cell cytokines such as IFN-γ, IL-2, and TNF-α [68]. CAF-derived EVs also led to shuttling of immunosuppressive PD-L1 into T24 cells. BALB/c nude mice implanted with subcutaneous T24 tumors and treated with CAF-derived EVs showed an increase in levels of *PD-L1* mRNA in BC cells as well as decreased CD8^+^ infiltration by immunohistochemical analysis [68]. This confirmed that CAFs can establish an immunosuppressive TME [68]. 

CAF-derived cytokines and chemokines can influence the immune cells in the TME [69]. Analysis of three TCGA databases showed a strong correlation between CAF-derived TGF-β, IL-6, and CXCL12 and an immunosuppressive TME in BC and poor response to immunotherapy [70]. Using two BC single-cell RNA sequencing datasets, iCAFs were found to associated with high infiltration of CD8 T-cells, regulatory T-cells, and M2 and exhausted T-cells [71]. In another analysis of the TCGA database, pathway analysis revealed CAF-associated calcium-binding protein caldesmon (CALD1) was associated with infiltration of M0 and M2 macrophages and was negatively associated with CD8 T-cell infiltration in BC [72]. Furthermore, the expression of T-cell exhaustion markers, such as CTLA-4, PD1, LAG-3, and TIM-3, was also associated with CALD1 expression, and CALD1 expression was correlated with the occurrence of high-grade tumors [72]. 

CAFs can influence the ECM through the deposition of structural proteins such as collagens and fibronectins, which impact the thickness and rigidity of the ECM. This in turn affects the recruitment and localization of tumor-infiltrating cells, the metastatic capacity of cancer cells, and the activity of cancer and non-cancer cells [73]. Interestingly, manipulation of the ECM by cancer cells has an important role in drug resistance, either through impairing drug penetration or through the activation of pro-survival signaling pathways in cancer cells [73]. CAF-derived ECM components such as collagens, elastin, and laminins; glycoproteins such as TNC, periostin, and fibronectin; as well as enzymes such as matrix metalloproteinases (MMPs) have been shown to shape the BC TME [74]. 

In bladder cancer, stromal cells including CAFs were shown to have high expression of various MMPs, especially MMP2, and the activation of MMP2 has been associated with TGF-β expression and activation of the p38 mitogen activated protein kinase (MAPK) pathway [75]. MMP2 has been implicated in the degradation of type IV collagen, thus increasing degradation of the ECM and facilitating BC migration and invasion. High expression of MMP2 has also been associated with increased angiogenesis [75]. Analysis of BC specimens from 106 patients revealed a negative correlation between diffuse TNC expression and overall survival [76]. TNC was observed in the stroma of both RT4 and RT112 organoids [77] as well as in human BC tissue samples [78]. Additionally, silencing TNC expression in the T24 and J82 BC cell lines reduced migration, invasion, and the expression of MMP2/9 in vitro [78]. 

Analysis of BC samples using the TCGA database highlighted the role of collagen type VI alpha 1 (COL6A1) as a hub gene, the expression of which was confirmed by immunohistochemical analysis of BC tissues and correlated with poor survival and response to anti-PD1 therapy [79]. BCG has also been shown to adhere strongly to the ECM structural protein fibronectin [80] and in murine bladders, treatment with anti-fibronectin antibodies impaired BCG attachment [80]. Thus, CAFs may influence the response to BCG therapy. Another CAF-derived ECM component, hyaluronic acid (HA), has been shown to influence the response to the chemotherapeutic drug mitomycin C [81]. NMIBC patients treated with a combination of hyaluronidase and mitomycin C showed better response compared to those who received mitomycin C alone, highlighting the tumor promoting role of HA [81]. 

Lastly, CAFs also influence the metabolic state of tumor cells, thus promoting survival in a hypoxic and acidic environment. CAFs produce and release lactate into the surrounding TME, which can be taken up by BC cells, thereby facilitating anaerobic respiration (also known as the Warburg effect) and promoting survival in the hypoxic TME [82]. CAFs express increased levels of monocarboxylate anion transporters (MCTs), which transport intracellular lactate into the TME. In a co-culture of CAFs and T24 cells, blocking of MCTs led to a decrease in secreted lactate concentration and increased T24 cell death [82]. 

## 4. Endothelial Cells/Pericytes

Tumor cells recruit endothelial cells (EC) and pericytes (PC) to the TME to induce the formation of blood vessels in order to support cancer cell growth and metastasis [83]. Unlike in healthy tissue, the tumor vasculature in BC is highly disorganized with leaky blood vessels that twist and bend around the tumor [83]. Single-cell analysis of bladder tumor samples from BC patients identified the presence of ECs that were positive for platelet endothelial cell adhesion molecule 1 (PECAM1), CD34, and VEGF receptor 1 (FLT1), and further clustering revealed five different subsets of ECs [84]. The bladder tumor tissue was enriched with VEGFR2^+^ and intracellular adhesion molecule 1 (ICAM1)high/atypical chemokine receptor 1-positive (ACKR1^+^) ECs, the presence of which was also associated with poor survival. VEGFR is the receptor for VEGF2 and KEGG analysis showed these VEGFR2^+^ ECs were associated with neovascularization pathways. The ICAM1^high^/ACKR1^+^ ECs were found to express pro-inflammatory cytokines TNF, IL-17, and nuclear factor kappa B (NF-kB) as well as chemokines CCL2, CXCL2, CXCL3, and CCL23, and they were also associated with the recruitment of myeloid cells into the TME [84]. Interrogation of the TCGA database to identify tumor-infiltrating ECs in BC [85] revealed that the EC cluster was associated with pathways involved in metabolism, such as fatty acid, arachidonic acid, and peroxisomal lipid metabolism [85]. Thus, a diversity of ECs can be found in the BC TME and may influence the TME through various pathways involved in inflammation and immune modulation.

ECs secrete various soluble factors through which they influence the surrounding TME. In vitro studies showed that activated HUVEC cells produced von Willebrand factor (VWF), which led to platelet aggregation that could increase blood vessel occlusion in BC patients [5]. Comparison of VWF levels in BC patients and healthy controls revealed higher blood VWF concentrations in BC patients, which was also correlated with tumor grade [5]. ECs were also shown to secrete epidermal growth factor (EGF), resulting in increased EGFR signaling (measured by increases in p-ERK, p-Akt and p-NF-κB protein levels) in T24 and 253J cell lines [17]. EC-derived EGF also increased the invasive capacity of T24 and 253J cells and the expression of EMT markers such as N-cadherin and zinc finger E-box-binding homeobox 1 protein (ZEB-1), which was abolished upon application of an EGFR inhibitor [17]. A similar phenomenon was observed in head and neck cancer, in which EC-derived EGF decreased the expression of epithelial markers (E-cadherin and desmoplakin), increased the expression of mesenchymal markers (vimentin and N-cadherin) [86], and increased the proliferation and self-renewal capacity of CSCs. The production of various cytokines and chemokines by ECs has also been reported; however, this has not been as well characterized in BC as in prostate cancer, where the EC-derived cytokines include IL-6, IL-8, IL-1, and TGF-β [87]. A head and neck cancer cell line (OSCC3) co-cultured with human ECs resulted in increased invasion and migration due to EC-induced expression of CXCL1 and CXCL8 [88]. 

Additionally, ECs may also be able to influence the TME to modulate the immune response. Single-cell sequencing of MCB6C bladder organoid tumors isolated from B6NTac mice treated with combined anti-PD-1/cytotoxic T lymphocyte associated protein 4 (CTLA-4) therapy led to the identification of an endothelial cell cluster positive for interferon gamma (IFN-γ) expression [89]. A repeat of the experiment in mice with IFN-γ receptor 1 (IFN-γR1) knocked out specifically in endothelial cells abolished the effect of the anti-PD-1/CTLA-4 therapy, and instead, tumors were found to grow rapidly at a rate comparable to the untreated mice, suggesting that EC-derived IFN-γ may be an important predictor of the response to immune checkpoint inhibitors [89]. ECs also influence the migration and extravasation of tumor cells and immune cell migration into the TME [90] through the expression of the adhesion cell surface molecule ICAM, which increased the interaction of ECs with T24 and J82 BC cells. Blocking ICAM resulted in decreased adhesion of both T24 and J82 to ECs [90]. 

Lastly, ECs also influence the metabolic state of tumor cells [91]. Compared to that of single-cultured T24 cells, those that were co-cultured with HUVEC cells (c-T24) had significantly higher expression of genes involved in oxidative phosphorylation, such as ubiquinol-cytochrome c oxidase reductase core protein 2 (*UQCRC2*), mitochondrially encoded cytochrome c oxidase subunit 1 (*MTCO1*), and ATP synthase 5A (*ATP5A*), and those involved in glycolysis, such as 6-phosphofructo-2-kinase/fructose-2,6-biphosphatase 3 (*PFKFB3*) and lactate dehydrogenase A (*LDHA*), compared to T24 cells cultured alone, suggesting that HUVEC cells may increase the energy consumption of BC cells [91]. Additionally, the expression of proliferation-related genes (proliferating cell nuclear antigen (*PCNA*), *cyclin D1*, *CDK5*, and *EGFR*) and migration rates were both higher in c-T24 cells. 

PCs also play an important role in the TME by supporting the formation of blood vessels, and they can be identified by the expression of markers such as platelet-derived growth factor receptor β (PDGFR-β), CD13, CD146, and α-SMA [92]. PCs can also secrete soluble factors through which they are able to influence the TME. For example, in a mouse melanoma model, tumor-derived PCs were found to express stimulatory molecules such as major histocompatibility complexes I and II (MHC I and II), CD80, and CD86, as well as immunosuppressive PD-L1 [93]. Such tumor-derived PCs impaired CD4^+^ T-cell function (measured by a decrease in the expression of activation markers) and induced T-cell anergy, highlighting their immune-modulating capacity [93]. PCs isolated from human glioblastoma biopsies were found to produce CCL5, which could promote tumor resistance to temozolomide in primary glioblastoma cells [94]. Single-cell sequencing of patient bladder tumors revealed the presence of PCs positive for regulator of G protein signaling 5 (PDGFRB^+^ PCs), which correlated with poor overall survival and resistance to atezolizumab therapy in BC patients [84]. However, further study is required to characterize pericytes in BC and to explore their influence on the TME. 

## 5. Other Stromal Cells (Adipose Cells/Adipose Stem Cells/Mesenchymal Stem Cells)

Obesity is a well-known risk factor in BC and is correlated with poor prognosis and an increased rate of recurrence [95]. Additionally, invasion of bladder tumors with perivesicle fat has also been correlated with poor survival [95]. Thus, adipose cells have gained traction as a tumor-influencing cell type present in the TME. Profiling the secretome of adipocytes obtained from bladder tumor samples using a human cytokine/chemokine panel revealed high levels of CCL3 (macrophage inflammatory protein-1α) and matrix metalloproteinase-9 (MMP9) [95]. Further analysis of the CM from bladder fat deposits revealed high levels of IL-8, growth regulated oncogene-alpha (GRO-α), and monocyte chemotactic protein-1 (MCP-1). Additionally, exposure of human T24 BC cells to the CM led to increased migration, as observed in a transwell migration assay [95]. High levels of cytokines (IL-6 and IL-8), chemokines (CCL2, CXCL1, and CX3CL1), and plasminogen activator inhibitor 1 (PAI1) were found in the CM and blocking of IL-6 or CXCL1 abolished the CM-driven migration of T24 cells [95].

However, studies have also revealed the anti-tumorigenic effects of adipocytes. A study by Kashiwagi et al. explored the effects of two adipocytokines (adiponectin and leptin) on UMUC3 and 647V BC cell lines [96]. Adiponectin was found to decrease migration and the expression of EMT-related proteins such as phospho-NF-κB, NF-κB, snail, slug, and COX-2. However, leptin was found to have the opposite effect on BC cells [96]. Adipose tissue stromal cells (ATSC) had differing effects on invasive and non-invasive BC cell lines [97]. Co-culture with ATSCs purified from the subcutaneous adipose tissue of Wister rats suppressed growth and increased apoptosis in RT4 cells (superficial BC cells), whereas the opposite effect was observed in EJ cells (invasive cell line) [97]. However, ATSCs increased the intracellular protein levels of MMP-2 and -9, phosphorylated rapidly accelerated fibrosarcoma (Raf), mitogen-activated protein kinase kinase (Mek), and extracellular signal regulated kinase (ERK), all of which are involved in proliferation and survival pathways, in both T24 and EJ cells [97]. Thus, the effect of adipocytes on BC cells may vary in a context-dependent manner and may depend on the status of the tumor-infiltrating adipocytes. A review by Wu et al. described the potential ability of tumor cells to influence tumor-infiltrating adipocytes to form cancer-associated adipocytes (CAAs), which have lower fat content and instead have higher expression of inflammatory signatures such as leptin, MMP-11, CCL2, CCL5, and IL-6 [98]. Whilst not well studied in BC, such CAAs would be able to influence both cancer cells and tumor-infiltrating immune cells. 

Adipose tissue can also give rise to adipose-derived stem cells (ADSCs) and mesenchymal stem cells (MSCs), and both ADSCs and MSCs are similar in nature and can both differentiate into adipogenic, myogenic, osteogenic, chondrogenic, and neurogenic cells [99]. ADSCs can be purified from the stromal vascular fraction of adipose tissue and have been found to suppress the proliferation of T24 and EJ cells when co-cultured in vitro [99]. The CM from ADSCs induced cell cycle arrest (increased cyclin-A and decreased CDK1 levels) and apoptosis (increased caspase 3 expression) in T24 cells [99]. Exposure of T24 and EJ cells to the CM from human ADSCs isolated from subcutaneous fat resulted in decreased proliferation, migration, and viability of both T24 and EJ cells [100]. Western blot analysis also revealed increased levels of the pro-apoptotic B-cell lymphoma protein 2 (Bcl-2)-associated X (Bax), whereas levels of the anti-apoptotic Bcl-2 was decreased in T24 cells following exposure to ADSC-derived CM [100]. Studies in CD133^+^ BC cells purified from HB-CLS-1 and 5637 cell lines and cultured in ADSC-derived CM showed an increased stem-like capacity, measured by an increased ability to form colonies [101]. Additionally, culture with ADSC-derived CM also increased the phosphorylation of AKT1/2/3 and ERK1/2 in CD133^+^ BC cells [101]. Thus, similar to that of adipocytes, studies have revealed differing and contrasting influences on the TME and BC cells. MSCs have also been shown to infiltrate the TME. MSCs can be derived from bone marrow as well as from the mesoderm, and single-cell sequencing analysis (of two databases) identified three sub-clusters of MSCs: cluster 0, which is enriched with immune-related genes; cluster 1, which is enriched with extracellular matrix-related genes; and cluster 2, which is enriched with adaptive immunity-related genes [102]. MSCs are thought to be recruited to the TME due to its inflammatory environment, especially due to the presence of chemokines that result in homing [103]. In a rabbit model, tumor-infiltrating MSCs were also found to differentiate into endothelial cells [103]. In the model, rabbit bladders were implanted with VX2 cells either with or without MSCs. The MSCs increased in vivo tumor growth and also increased intra-tumoral expression of bFGF, TGFβ1, hepatocyte growth factor (HGF), MMP2, and MMP9 [104]. In a follow-up study, blocking the TGFβ1 receptor or Smad2 abolished the MSC-driven increase in tumor growth [105]. The TGFβ1-Smad signaling pathway has been shown to be involved in the recruitment of immune cells, such as MDSCs and regulatory T-cells, as well as activation of a suppressive TME. MSCs also release EVs, which can influence the TME. EVs from human MSCs isolated from adipose tissue decreased the viability and proliferation of 5637 BC cells in vitro and also exerted anti-tumor effects by increasing the expression of p53 in 5637 cells [106]. The interaction between the various cell types is shown in Figure 2. 

## 6. Do Neurons Support Cancer Growth and/or Inhibit It?

Bladder expansion and voiding requires the coordinated effort of nerves and muscles. The predominant nerves innervating the dome and lateral walls of the bladder are derived from the spinal S2–S4 nerves via the pelvic splanchnic nerves, while the posterior wall is predominantly innervated via the hypogastric nerves from the sympathetic lowest thoracic and lumbar splanchnic nerves. In rabbits, skin denervation increased chemical-induced tumor growth, as demonstrated by tumor growth in the denervated versus normal ear [107]. Meanwhile sympathetic nerve denervation was shown to slow the growth of breast cancer cell lines [108]. These studies indicate that nerves have some measure of control of tumors, but the best evidence of tumor interaction with nerves has come from the perineural spread of tumors. While well known in other cancers, evidence is not as extensive in bladder cancer. Perineural spread for bladder tumor cells has been suggested to explain the location of tumor cells in or near nerves without location of the tumor anywhere nearby [109]. Lumbosacral plexopathy occurs in about 2% of patients. Based on studies in other cancers, it appears that the vagus nerve supports tumorigenesis via cholinergic receptor activation and sympathetic nerve activation or inhibition promotes pro or anti-tumor effects [110]. In prostate cancer, it was shown that the stroma contained double cortin-expressing (DCX^+^) neural progenitor cells that promote cancer growth [111]. Normally, DCX cells are found in adult neurogenic areas of the central nervous system (CNS) and the new neurons they generate move to sites of injury. 

Cancer cells also secrete factors that attract axons. The interaction of nerves with cancer cells may occur via a synaptic junction or by the secretion of neurotransmitters. Cancer cells are known to express neuronal cell adhesion molecules, neuron restrictive silencing factors and neurotransmitters, receptors for neuronal factors, and voltage-gated ion channels. Three neuron-associated proteins were examined in bladder cancer tissue. Synculeins are small soluble proteins expressed on neural tissue and tumors. Ninjurin is produced during nerve injury and is necessary for nerve growth. It is expressed on epithelial cells and is considered a cell-adhesion molecule. Neuropilins are transmembrane receptors that bind semophorins and VEGF and thus regulate neurons and angiogenesis. Synculeins were strongly expressed in advanced-stage tumors of the bladder and ninjurin expression was predictive of tumor progression on univariate analysis [112]. In the tumor microenvironment, neural progenitor cells and sensory neurons can differentiate into adrenal cells producing catecholamines. 

Brain-derived neurotrophic factor (BDNF) binds to tropomyosin receptor kinase B (TrkB) and regulates neuron differentiation and maturation. TrkB is expressed on normal urothelial cells and at a higher level on transitional cell carcinoma (TCC) cells [113]. Suppressing TRkB activation with miR-1-3p mimics, which reduced BNDF levels, induced cytotoxicity [18]. Weekly injection of BNDF increased xenograft growth in SCID mice and this was reduced by injecting chimeric TrkB-Fc protein, thus confirming the action of these proteins on tumors [114]. The impact of BDNF secretion by colon tumor cells has also been shown to correlate with neurite growth into tumors [115].

Adrenergic (G protein-coupled) receptors are receptors for noradrenalin and epinephrine (catecholamines) from the autonomic sympathetic nervous system and the adrenal medulla. Alpha-1 and -2 adrenoreceptors (α1 and α2 ARs) and β-1, -2, and -3 adrenoreceptors (β1, β2, and β3 ARs) oppose each other, the former mediating smooth muscle contraction and vasoconstriction and the latter mediating relaxation and vasodilation. Binding of catecholamines by α1 AR triggers the expression of Gq G proteins, causing the hydrolysis of membrane phospholipids and generating inositol phosphate (IP) and diacylglycerol (DAG). The former mobilizes calcium from intracellular stores and causes smooth muscle contraction. α2 AR opposes α1 AR and acts via the Gi G proteins to inhibit adenylate cyclase, which in turn inhibits norepinephrine release from presynaptic neurons. In turn, α1 AR activation blocks the production of new α2 AR. α1 AR can drive proinflammatory cytokine production and α1 receptor agonists cause the vasodilation of blood vessels and suppress excitatory nerve signaling in the urinary tract. 

The α1D AR antagonist, naftopidil, suppressed the excitatory effects of bladder distension and prolonged the inter-contraction interval of bladder cells. Clinically, α1 AR antagonists are used to treat hypertension, but several, like the quinazoline-derived antagonists, are able to block the growth of bladder cancer cell lines. Naftopidil induced apoptosis via activation of caspase-3 in bladder cancer cells and reduced xenograft tumor growth in vivo [116,117]. The α1 AR antagonist phentolamine also blocked angiogenesis [118]. Taking α1 blockers was shown to reduce the incidence of bladder cancer in a retrospective observational study of 27,813 male patients (some of whom were prescribed α1 AR blockers for hypertension or benign prostate hyperplasia). The cumulative incidence of developing BC was 0.24% compared to 0.42% in the group that did not take α1 blockers [119]. This showed a chemo-preventive effect of medication on bladder cancer development. Evidence of an impact on disease progression comes from an immunohistochemical examination of tissues obtained by radical cystectomy, which showed reduced microvessel density and increased apoptosis in tissues from those who had long-term exposure to the α1 AR blocker terazosin (a quinazolone derivative). One caveat was that only 21 samples were examined (9 exposed to the blocker and 13 without) [120]. Silodosin, another α1 AR, was also found to affect bladder cancer cells and also correlated with better progression-free survival in NMIBC patients [121]. 

Similarly, β AR antagonists have anticancer effects. Binding of catecholamines by β AR on cancer cells leads to cellular changes that favor cytoskeletal remodeling, generating invasive protrusions and ECM-degrading proteases. Meanwhile, β adrenergic signaling can also impair DNA repair mechanisms, leading to genomic instability in tumor cells [122]. β AR signaling enhances tissue vascularity and promotes angiogenesis [123]. In a Swedish study of 16,669 subjects, beta blockers were associated with overall lower bladder cancer-specific mortality [124]. Anti-tumor effects of beta blockers were demonstrated on bladder cancer cell lines treated with the beta-blocker propranolol. Propranolol exhibited anti-proliferative and pro-apoptotic effects in vitro and in vivo on mouse xenografts by modulating Na^+^/H^+^ exchange as well acting on immune cells to activate a systemic immune response [125].

## 7. Impact of Anti-Hypertensive Drugs and Anesthetics on Bladder Cancer

The renin-angiotensin-aldosterone system (RAAS) regulates blood pressure and volume. Renin catalyzes the production of angiotensin II (Ang II) from angiotensin and Ang II acts on sympathetic nerves to enhance norepinephrine production. Ang II and Ang III bind to the receptors AT1R and AT2R. Anti-hypertensive drugs include ACE inhibitors (ramipril, lisinopril, and enalapril), receptor-binding blockers (irbesartan, valsartan, losartan, and candesartan), calcium channel blockers, and beta blockers. Based on a meta-analysis of 31 articles analyzing approximately 3 million participants, only angiotensin II receptor blockers (ARB) showed a significant association with the development of bladder cancer [126].

However, bladder cancer patients with CIS (n = 64) who were taking antihypertensive drugs and received BCG immunotherapy had better recurrence-free survival (RFS), especially in the group with gross hematuria [127]. Analysis of the use of angiotensin receptor (ATR) blockers in 14,065 Finnish subjects prior to being diagnosed with BC found that it was associated with a slightly decreased risk of death (HR = 0.81, CI = 0.71–0.93) and the association was dose dependent [128]. Post-diagnostic use of ATR-blockers, when compared to non-use, was similarly associated with better survival [128]. Yoshida et al. also found in an analysis of patients (n = 269) undergoing radical cystectomy that those taking RAS inhibitors had better overall survival, with a 5-year survival of 79% versus 66.4% for those not taking RAS inhibitors, [129].

Interestingly, use of calcium channel blockers was also associated with better survival and the risk of bladder cancer death decreased with increasing intensity of use (HR = 0.067, CI = 0.52–0.86 for highest intensity) [128]. A single-center analysis revealed that high-risk NMIBC subjects who were taking ARB but not ACE-I had better outcomes when receiving BCG immunotherapy (6 instillations followed by monthly instillations of up to 12 months) [130]. In contrast, Blute Jr. et al. evaluated 340 patients who underwent TURBT, received BCG therapy, and were on ACE-I/ARB therapy, finding that those who received BCG and ACE-I/ARB therapies had higher RFS than those who received BCG therapy alone, based on multivariate analysis [131]. Shen et al. [132] performed a meta-analysis and found that RAAS inhibitors also improved the response to immune checkpoint inhibitors (ICIs) in patients with several cancers, including melanoma, NSCLC, and urothelial carcinoma [133].

Decisions on the type of anesthetic and their route of delivery have been reported to have some impact on the outcomes of surgery in multiple cancer types [134]. Inhalant anesthetics (general anesthesia, GA) can generate a pro-tumorigenic atmosphere by increasing levels of circulating VEGFC [135] as well as cause immunosuppression [136]. Surgery can release cancer cells into circulation and the local environment, which could lead to metastasis. The trauma associated with surgery can induce stress via the hypothalamic pituitary axis and the sympathetic nervous system, causing the release of catecholamines, glucocorticoids, prostaglandins, and endogenous opioids, all of which promote immunosuppression and enable the cancer cells to survive. For example, μ opioid receptor agonists used to reduce pain during and post-surgery have been shown to increase circulating tumor cells in mice [137].

Local anesthesia includes intravenous (IV), regional anesthesia (RA), peripheral nerve block, epidural, and spinal anesthesia (SA). In epidural and nerve block, a lot of anesthetic enters circulation and may directly interact with cancer cells in circulation or at the tumor site. RA is supposed to block afferent signals from surgery from being sent to the central nervous system, decreasing activation of sympathetic nervous system, endogenous opioid release, and inflammation. In vitro, RA drugs have been shown to impact cancer cell viability [138]. Li et al. examined the effects of propofol, a commonly used anesthetic, on BC cells and showed it affected cell proliferation, migration, and stem cell self-renewal at the concentrations used clinically [139]. Bupivacaine (anesthetic)-induced apoptosis and ferropotosis in bladder cancer cells lines and i.p. injection of the drug reduced the growth of bladder cancer xenografts in mice. The effect was due to ferropotosis, which occurred due to reduced cellular GSH levels and increasing lipid peroxidation, blocking x-CT (glutamic acid/cystine transporter), which caused Fe++ accumulation, glutathione peroxidase 4 gene expression, and suppressed phosphorylation of the PI3K/AKT/mTOR pathway [140].

Several retrospective studies on NMIBC outcomes with respect to SA versus GA have reported contradictory outcomes. Two studies showed reduced recurrence with SA rather than GA. Choi et al. examined outcomes for 876 patients, 718 of whom received SA and 158 received GA, and reported at 5 years that the recurrence rate was lower in the SA group [141]. Koumpan et al. analyzed 231 patients (n = 135, GA and n = 96, SA) and found lower recurrence with SA [142]. In contrast, Lee et al. examined 4349 patients, 3767 of whom received GA and 582 received RA, and found at a follow-up of more 5 years that the method of anesthesia delivery did not affect recurrence [143]. A meta-analysis of 8 studies with a total of 3764 patients with NMIBC, 2117 of whom received SA and 1647 received GA, found that those who were treated with SA had better outcomes [144]. 

A retrospective study by Pfail et al. found that in robot-assisted radical cystectomy for high-risk NMIBC and MIBC, those given GA such as sevoflurane, isoflurane, and desflurane (n = 105) or intravenous anesthetic propofol (n = 126) were more prone to recurrence than those treated with intravenous anesthetics [145]. Chipollini et al. questioned whether the outcomes may have been related to the type of anesthetic used, as they reported poorer outcomes for patients undergoing radical cystectomy who were given the epidural sufentanil [146]. In a study on NMIBC comparing the use of GA (propofol, gentanyl, cis-atracurium, remifentanil, and sevoflurane) versus LA (lidocaine, ropivacaine, and bupivacaine), 662 of whom received GA and 264 received LA, no difference was found in recurrence or progression between the groups. But after propensity score matching to account for confounding differences, there was a progression-free survival difference between the groups (*p* = 0.036) [147]. Consistent with this belief, a prospective study of 100 patients undergoing radical cystectomy who received GA versus GA plus LA reported lower disease-free survival for the latter combination where patients received propofol + LA versus sevoflurane + opioids [148], but their follow-up period was short. Hopefully, new studies can clarify this point. A new trial on NMIBC (NCT03597087) was initiated to investigate the impact of LAN on cancer recurrence. The oral drug 5-aminolevulinic acid used to help visualize tumors during TURBT was also found to cause hypotension, which in turn could impact surgical outcomes in terms of tumor recurrence [149].

## 8. Conclusions

The knowledge of the TME that has been revealed by single-cell analysis and spatial biology has opened new vistas in our understanding of tumors, as shown in Figure 2. Further, this has also revealed the complexity of tumors and that a failure to respond to therapy may occur for very different reasons in different patients. Given this complexity, it is thus quite surprising when monotherapy works well. This may very well be related to factors not previously considered that could have influenced outcomes, such as the drugs patients are taking for other health problems. This review also highlights that treating only the cancer may not be sufficient. Rather, the patient’s health status in terms of concurrent medication and even anesthetics used at the time of surgery to remove the tumor all could impact the cancer cells and TME. A more holistic approach is needed to treat cancer.

## 9. Future Direction

Bladder cancer more than other cancers generally occurs more frequently in older people, and as such, consideration should be given to the impact of age on treatment and therapeutic strategies. The choice of anesthesia and route of administration is often impacted by patient comorbidities and age. Thus, the design of future clinical trials to evaluate therapies or surgical strategies should address this point in their design. Further, the role of the microbiome, both the bladder and gut microbiomes, in the response to therapy needs to be evaluated as well, as it may provide more avenues to improve patient outcomes.

## Figures and Tables

**Figure 1 ijms-24-12311-f001:**
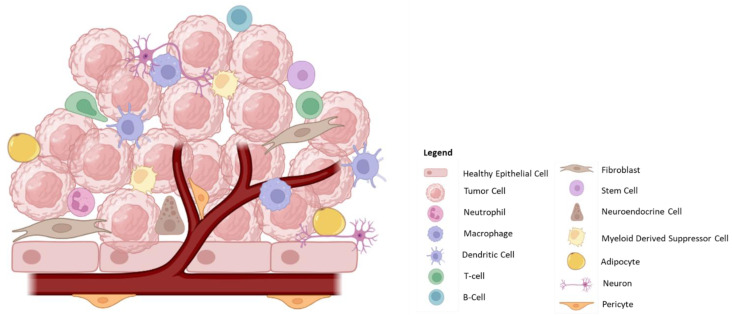
The Tumor Microenvironment.

**Figure 2 ijms-24-12311-f002:**
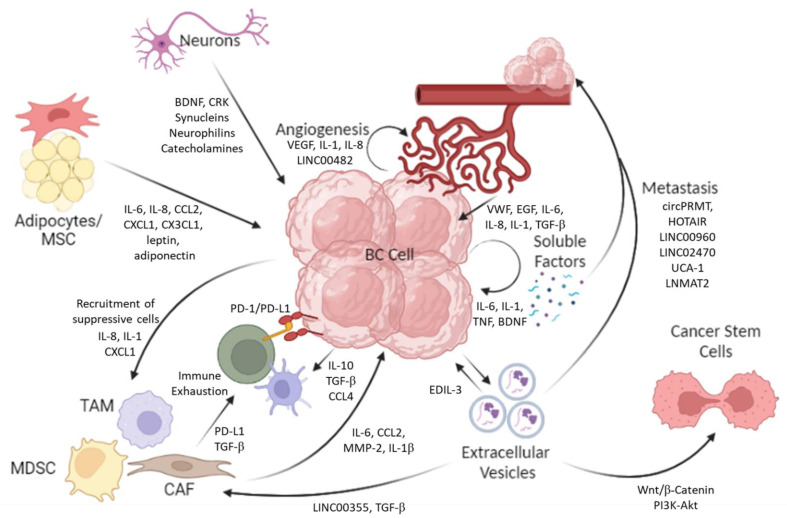
The dynamic action of different cell types that produce the mature TME.The diverse cells present in the TME influence one another, which contributes to tumor growth and prognosis. The outgoing arrows indicate the release of soluble molecules from different cells that impact the target cells within the TME. BC cells propagate their own proliferation and migration (metastasis) through autocrine signaling pathways, involving the production of tumor-promoting and pro-inflammatory cytokines such as IL-6, TNF, and IL-1. BC cells secrete EVs, which can further activate tumor cells in a positive loop and induce tumor cell metastasis, especially through the delivery of non-coding RNAs. Tumor-derived EVs also fuel an immunosuppressive environment by recruiting and activating tumor-infiltrating fibroblasts. These CAFs and pro-tumorigenic immune cells, such as MDSCs and TAMs, are recruited to the TME by the cytokines and chemokines secreted by the BC cells. Once in the TME, these cells provide feedback to the tumor cells by secreting soluble molecules and also inactivate other immune cells, such as T-cells, by upregulating the expression of exhaustion markers. BC cells also recruit and activate endothelial cells (angiogenesis) through the secretion of soluble factors such as IL-1, IL-8, and VEGF, as well as long intergenic non-coding RNAs such as LINC00482. These activated endothelial cells form disorganised networks of blood vessels, which are leaky and supply the tumor cells with life-sustaining nutrients. Endothelial cells, adipocytes, and neurons, in turn, influence the tumor cells by secreting inflammatory cytokines, chemokines, growth factors, hormones, and other signaling peptides.

**Table 1 ijms-24-12311-t001:** Cell types present in the TME.

Cell Type	Cell	Tumorigenic Potential	Description/Properties
Healthy Tissue	Epithelial Cells		Healthy Cells
Tumor	Cancer cells	+	Dysregulated, rapidly dividing cells that establish the tumor microenvironment to allow for growth and metastasis.
	Cancer Stem Cells	+	Stem-like cells with the potential to rapidly divide in order to restore the cancer cell population. These cells have been implicated in tumor recurrence.
Immune	Neutrophils	+/−	Pro-inflammatory response involving the generation of cytokines/chemokines and reactive oxygen species (ROS), which promotes tumor cell death; also important in recruitment of T-cells. However, in late-stage tumors, neutrophils promote angiogenesis and metastasis.
	Tumor-Associated Macrophages (TAM)	+/−	Can be either pro-(M2) or anti-tumorigenic (M1). Tumor-resident macrophages are highly important in regulating the surrounding environment by the generation of inflammatory molecules, cytokines/chemokines, ligands, and through interaction with infiltrating immune cells.
	Dendritic Cells (DC)	−	Important in antigen presentation and activation of T-cells within the tumor microenvironment. However, DCs are susceptible to immunosuppressive environments.
	NK Cells	−	Low penetration into the tumor microenvironment but are potent in lysing tumor cells.
	Myeloid-Derived Suppressor Cells (MDSCs)	+	Exert an immunosuppressive effect through the production of cytokines/chemokines.
	T-cells	−	Highly important component of the anti-tumorigenic response. Mediate tumor cell death.
	Regulatory T-cells	+	Contribute to an immunosuppressive environment resulting in T-cell inactivation/exhaustion, which promotes tumor growth.
	B-cells	+/−	Important in antigen presentation and antibody generation; however, the presence of regulatory B-cells in some tumors contributes to immune suppression.
Stromal Cells	Endothelial Cells	+	Important for angiogenesis and nutrient supply to the tumor. Tumor-derived endothelial cells also present with altered morphology and function, forming leaky and disorganized vasculature.
	Pericytes	+/−	Support angiogenesis and may promote tumor metastasis. However, pericytes may also aid in the recruitment and activation of immune cells.
	Cancer-associated Fibroblasts (CAFs)	+/−	Largely studied for their pro-tumorigenic functions due to their involvement in creating an immunosuppressive environment and further supporting tumor cells by supplying nutrients. Also heavily involved in the generation of the extracellular matrix.
	Adipose Cells	+	Regulate the tumor microenvironment by secreting nutrients, growth factors, hormones, peptides, enzymes and cytokines.
Innervation	Neurons/Neuro-endocrine Cells	+	Secrete signaling molecules, ligands, and peptides, which promote tumor growth and proliferation and intra-tumoral signaling.

“+” protumorigenic potential, “−” no tumorigenic potential.

## Data Availability

Not applicable.

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
