# Peer review of "The Bladder Tumor Microenvironment Components That Modulate the Tumor and Impact Therapy"

_ijms, 2023, doi:10.3390/ijms241512311_

Round 1
Reviewer 1 Report
The review is interesting, however needs to be written well and restructured before accepting it. 1. Change the heading to Bladder cancer cells as it talks only about bladder cancer cells. 2. Expand the abbreviation when using it for the first time even though it sounds common. ex: IL-1 , IL-1ra, TIMP-1, VEGF-A etc 3. Line 52- What is LINC000482? 4. lIne 55 - cancer cells also secrete means? compared to what? 5. Line 65- What is CXCL1? 6. Line 91- Rewrite sentence 7. Line 212 - remove the word bladder in the sentence to correct it. 8. Line 246 - what is LINC00355? 9. Line 399- In this study bladders were implanted? 10. Both Figure 1 and 2 need more description and explain outgoing and incoming arrow marks with respect to BC cells in figure 2. 11. Does Ephrin B1 belong to the exosomes section? doesn't fit in the section and sentences are redundant . 12. Line 499- section 4- hypertensive or antihypertensive ? 13. Remove capital letters in between sentences in line 519 14. What is TURBT? 15. What is SA? use abbreviation in bracket after the expansion for the first time 16. rewrite et al as et al., through the manuscript 17. Line-549 What is the meaning of this sentence? 18. Line 551- Start sentence with capital letter (anesthesia) 19. Future directions is missing.
Minor editing of English language required.
Author Response
Replies are highlighted in blue in the manuscript
Point 1. Changed heading to “Bladder cancer cells”
Point 2 and 3 and 5. Expanded abbreviations. This has been performed throughout the manuscript. Table 1, line 37, 39, line 41, lines 47 and 48, 54, 55, 71, 72
Point 4 line 55 has been changed.
Point 6 the sentence has been rewritten.
Point 7 line was corrected as requested.
Point 8 LINC0035 was explained. It is a long non coding intergenic RNA.
Point 9 the study was referring to patient outcomes.
Point 10 The captions were included for Figure 1 and 2.
Point 11. Ephrin section has been moved.
Point 12. Corrected to read as Anti-hypertensive
Point 13 sentence was corrected.
Point 14 TURBT was explained on the first page, it is transurethral resection of the bladder tumor
Point 15 SA has been explained. It is spinal anesthesia.
Point 16 “et al., “ has been corrected
Point17 This refers to a meta- analysis that looked at several retrospective studies comparing patient outcomes after surgery depending on whether the patients received spinal or general anesthesia. They reported that they saw better outcomes for those whose received Spinal anesthesia. IT has been rephrased and is now line 664
Point 18 Corrected the sentence
Point 19 future directions included.
We rephrased some parts of the manuscript and these are highlighted in grey.
Reviewer 2 Report
Summary:
In this review article, Patwardhan & Mahendran describe in detail the different cell types (cancer cells, cancer stem cells, endothelial cells, pericytes, adipose cells, cancer associated fibroblasts and neuronal cells) in the bladder tumor environment. In addition, the authors describe their impact on immune activation, as well as on shaping the environment. They also discuss the role of hypertensive drugs and anesthetics on bladder cancer. Overall, the manuscript is very well organized on relevant topics. However, I do have some suggestions that might help improve the quality of the paper.
- Throughout the manuscript, the authors cite different molecular pathways, as well as numerous signals capable of activating the epithelial-to-mesenchymal transition process. In my opinion, the authors could describe, even briefly, the concept of EMT process. This is important for lay readers on the subject.
- I suggest the authors revisit topic #4. "Impact of hypertensive drugs and anesthetics on bladder cancer". This subject has been extensively explored in recent years, and there are still many discussions in the literature on this very relevant topic. Recently, new articles were published, and could be inserted in this topic (PMID: 35982423, PMID: 33219885, PMID: 33522306). I believe this would make the manuscript more attractive to readers.
- The authors describe the importance of CAFs in the TME, and give more attention to the production of MMPs and degradation of ECM components. However, it is already known that CAFs also secrete other ECM components capable of modulating the immune response, as well as the drug resistance phenotype. I suggest the authors describe these matters of great relevance in oncobiology in the BC context.
- I suggest the authors wrote the captions for figures 1 and 2. The figures are beautiful and carry a lot of information. However its captions are poor in content.
Author Response
We thank the reviewer for his suggestions.
All replies are highlighted in yellow in the manuscript.
We have included information on CAF effects on the EC.
We have also explained EMT.
A few more references with respect to anesthesia and outcomes for bladder cancer surgery were included as suggested.
The captions for Figure 1 and 2 have been included.
Reviewer 3 Report
This review is a very good summary of the topic regarding the components of the bladder tumor microenvironment (TME). Based on the current knowledge the authors discussed how various cell types in the TME impact the developing cancer. The most important information was well summarized in figures and tables. In vitro, to a lesser extent in vivo, studies were detailed in the text. The work is well-thought, well-organized, and includes most published articles. Generally, I have not found significant limitations in this manuscript; conversely, I think that it has many strengths, such as an interesting theme and an accurate presentation of the issue.
I have only a few minor remarks:
- In the Introduction section, I would add a few sentences about the problems of bladder cancer treatment, which would better justify the topic.
- Line 33: I think there should be any reference(s) here.
- Lines 152, 473, 495: ‘in-vivo’ – be consistent. Check also ‘et al.’ in the whole text.
- There are some editing errors like double spaces that should be corrected.
English is fine, only minor editing of English language is required.
Author Response
We thank the reviewer for his comments.
All replies in the manuscript are highlighted in green.
Information about bladder cancer and problems with treatment have been included in the introduction and the references added.
Corrections were made in the text for et al., and in-vivo.
Round 2
Reviewer 2 Report
All my concerns have been addressed